# Only a minority of bacteria grow after wetting in both natural and post-mining biocrusts in a hyperarid phosphate mine

Talia Gabay[1,2], Eva Petrova[3], Osnat Gillor[2], Yaron Ziv[1], and Roey Angel[3*]

[1]Department of Life Sciences, Ben Gurion University of the Negev, 8410501, Israel

[2]Zuckerberg Institute for Water Research, Blaustein Institutes for Desert Research, Ben-Gurion University of the Negev, 8499000, Israel

[3]Institute of Soil Biology and Biogeochemistry, Biology Centre CAS, Na Sádkách 7, 370 05 České Budějovice, Czech Republic

Correspondence: Roey Angel (roey.angel@bc.cas.cz ), Talia Gabay (taliajoann@gmail.com)

## Abstract

Biological soil crusts (biocrusts) are key contributors to desert ecosystem functions; therefore, biocrust restoration following mechanical disturbances is imperative. In the Negev Desert hyperarid regions, phosphate mining has been practiced for over 60 years, destroying soil habitats, and fragmenting the landscape. In this study, we selected one mining site restored in 2007, and used DNA stable isotope probing (DNA-SIP) to identify which bacteria grow in post-mining and adjacent natural biocrusts. Since biocrust communities activate only after wetting, we incubated the biocrusts with $H_2^{18}O$ for 96 hours under ambient conditions. We then evaluated the physicochemical soil properties, chlorophyll *a* concentrations, activation, and functional potential of the biocrusts. The DNA-SIP assay revealed low bacterial activity in both plot types and no significant differences in the proliferated communities' composition when comparing post-mining and natural biocrusts. We further found no significant differences in the microbial functional potential, photosynthetic rates, or soil properties. Our results suggest that growth of hyperarid biocrust bacteria after wetting is minimal. We hypothesize that due to the harsh climatic conditions, during wetting bacteria devote their meager resources to prepare for the coming drought, by focusing on damage repair, and organic compound synthesis and storage rather than on growth. These low growth rates contribute to the sluggish recovery of desert biocrusts following major disturbances such as mining. Therefore, our findings highlight the need for implementing active restoration practices following mining.

## Keywords

Biological soil crust; Biocrust restoration; Stable isotope probing; Hyperarid desert; Mining;

Ecological restoration

38

39

# 1. Introduction

Phosphate mining in the Negev Desert, Israel, has been taking place since the 1960s in large areas. ILC-Rotem mining company leads the phosphate mining activities and has been practicing a reclamation-oriented mining protocol for the past 15 years. The mining protocol entails the excavation of the top 50-70 cm of soil (which they consider to be topsoil) followed by the overburden (the layer covering the phosphate), then storing the two soil layers in separate piles.  Following the excavation of the phosphate, the overburden is returned to the mining pit followed by the topsoil. Finally, the terrain is leveled with heavy machinery. The area is then considered a restored, post-mining site.

Open mining activities lead to the destruction of the local vegetation and seed bank, and the fragmentation of the natural landscape (Sengupta, 2021). The consequences include land degradation, erosion, soil and water pollution, and dust dispersion. In addition, mining activity often leads to decreased biodiversity in and around mining sites (Bridge, 2004, Sengupta, 2021). One of the ecosystem components being destroyed by mining activities in the Negev Desert is the biological soil crust layer (biocrust). Biocrust is the topmost layer of many arid soils and comprises primary-producing and heterotrophic microorganisms that bind together soil particles using secreted extracellular polymeric substances (EPS), mainly polysaccharides (Weber et al., 2022). Biocrusts provide many ecosystem services, including fixing nitrogen and carbon, and soil stabilization (Belnap and Lange, 2003). While biocrust microorganisms developed various adaptations to withstand the harsh desert environment

(Makhalanyane et al., 2015), biocrust structures are susceptible to mechanical disturbances.
Such a disturbance, especially over large scales (for example, mining activity), breaks and
buries biocrust organisms, often resulting in changed biocrust communities (Belnap and
Eldridge, 2003).

In a previous research, we evaluated the biocrust bacterial communities in phosphate
mining sites (Gabay et al., 2022). Briefly, we found that natural and post-mining biocrusts
differ in community composition and diversity. Following the biocrust community analysis,
we sought to identify which bacterial groups are actively growing in the biocrust and
whether the composition differs between natural and post-mining sites. To this end, we used
DNA-stable isotope probing (DNA-SIP): a culture-free approach that allows the detection of
actively growing microorganisms by labeling them with stable isotopes such as $^{15}N$, $^{14}C$, and
$^{18}O$ (Dumont and Hernández García, 2019). SIP has been widely applied in identifying
microbial groups that participate in carbon and nitrogen cycling, such as methanotrophs
(Sultana et al., 2019, Zhang et al., 2020), methylotrophs (Macey et al., 2020, Arslan et al.,
2022), and nitrogen fixers (Pepe-Ranney et al., 2016, Angel et al., 2018). Likewise, SIP can use
the incorporation of heavy water ($H_2{}^{18}O$) into various biomarkers to study the growth and
function of microorganisms that become activated upon wetting (Schwartz et al., 2019).
Previous $H_2{}^{18}O$ SIP experiments measured microbial growth rates and dynamics following
hydration (Blazewicz et al., 2020). Desert biocrusts make an ideal study system for $H_2{}^{18}O$ SIP
experiments, as they become active quickly following hydration (Angel and Conrad, 2013),
resuming growth, nutrient cycling, and excretion of extracellular organic materials (Garcia-
Pichel and Belnap, 1996, Belnap and Lange, 2003).

In this research, we investigated the proliferation of bacterial groups in biocrusts taken from
reference ('natural') areas and post-mining sites by incubating biocrust samples with
isotopically-labeled water ($H_2^{18}O$). We hypothesized that growth patterns and taxonomic
identity of bacterial groups would differ significantly when comparing natural and post-
mining biocrusts. Specifically, we expected higher bacterial growth rates in natural
compared to post-mining biocrusts. Based on our previous findings, we specifically expected
higher activity of Cyanobacteria in the natural biocrusts (Gabay et al., 2022).

## 2. Materials and Methods

### 2.1. Study site and sample collection

Sampling was conducted during June 2020 at the Gov Mining Site, located in the Zin Valley (30.84 °N, 35.09 °E, 98 m above sea level), where restoration was completed in 2007. The study area was previously described in Gabay et al., (2022). Briefly, Zin Valley is a hyperarid region of the Negev Desert, with 50 mm average annual rainfall (Zin factory meteorological data) and highly saline soils (average EC = 24 dS/m). The main soil cover types in Zin Valley are biocrusts and desert pavement, with scarce vegetation of mainly annual species. The soil composition in the post-mining site and natural area is similar with 70% sand, 18% silt and 12% clay, and 68% sand, 20% silt and 12% clay for natural and post-mining respectively (Gabay et al., 2022). The soils in Zin Valley are classified as Solonchaks according to the World Reference Based soil classification system.

Biocrusts were sampled either from the post-mining site or the adjacent natural area. The biocrusts in Gov are thin (between 1.5 - 2 .5 mm deep), and smooth. The site is characterized by areas covered in biocrusts or desert pavement. In each sampling site, we sampled along a 100 m strip at approximately 10 m intervals (Fig. 1). In total, we sampled 20 biocrust samples (10 from each site). We collected the biocrusts using a spatula, at an average depth of 2 mm. Biocrusts were placed in 100 mm x 15 mm petri dishes lined with cotton. For the SIP assay, we

chose 5 of the 10 samples from each site containing the highest chlorophyll *a* concentrations
as estimated in preliminary experiments (Table S1).

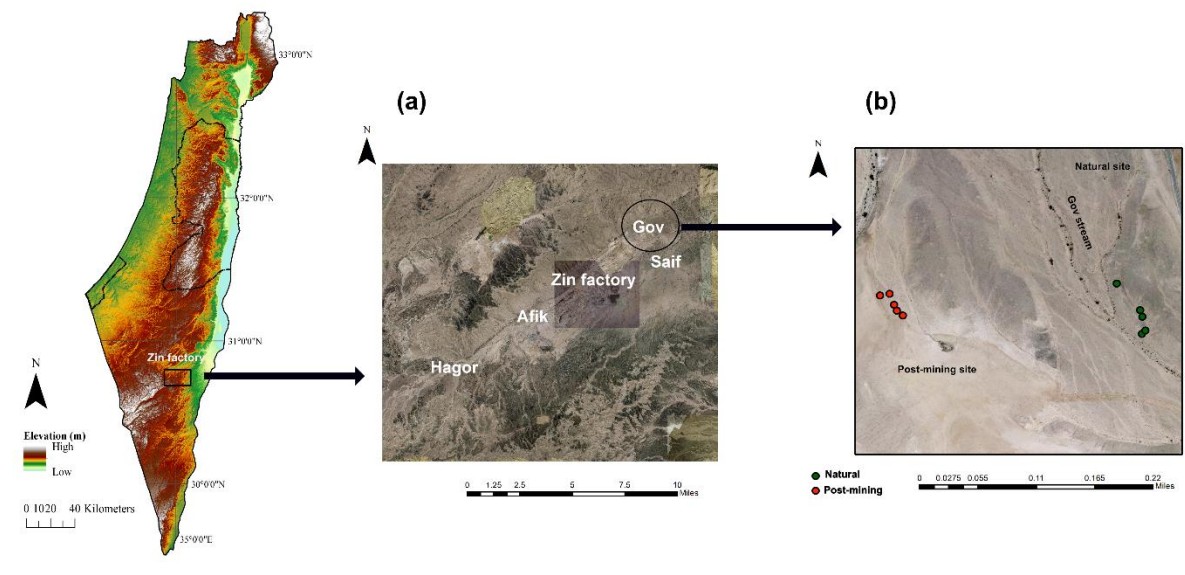


Figure 1: map of the research area. Map a shows the different post-mining sites around the Zin
factory. Map b shows the biocrust sampling points in gov mining site used for this research.
Green dots represent the natural biocrusts, and red dots represent the post-mining biocrusts.
## 2.2. Soil properties
Five biocrust samples from each plot type (post-mining and natural) were sent for analysis of
soil properties (pH, EC, $NO_3^-$ concentrations, and soil organic matter). The analysis was
performed at the Gilat Soil Laboratory (Gilat Research Center, Gilat, Israel).

2.3. chlorophyll *a* extraction
Chlorophyll *a* was extracted from biocrust samples using a protocol previously described in
Gabay et al., (2022). Briefly, chlorophyll *a* was extracted from 3 g soil of each biocrust sample
was diluted in 9 mL of methanol for 15 min at 65 °C. The soil solution was centrifuged at 2000
rpm for 5 minutes, supernatant was collected, and absorbance was measured in a
spectrophotometer at 665 nm. Concentrations were calculated according to (Ritchie, 2006)
and normalized to 1 g of soil. Extractions of the biocrusts were performed before (dry
biocrusts) and after 96 hr incubation with distilled water (DW) under identical conditions to
the incubation with $H_2^{18}O$.

## 2.4. Stable isotope probing

### 2.4.1. Soil incubation

To test the incorporation of $^{18}O$ into biocrust samples, a microcosm was designed to control
for the incubation conditions. Each microcosm consisted of a 10 mL glass vial in which 1 g of
biocrust sample was placed. To achieve field water-holding capacity, 0.15 mL of $H_2^{18}O$ or
DNase-free water were added. The glass vials were then sealed with butyl rubber stoppers
(Sigma-Aldrich, St. Louis, Missouri, United States) to prevent evaporation. Both labeled and
unlabeled controls were incubated in duplicates, for a total of 40 vials. Samples were
incubated under a 12 hr photoperiod for 96 hr in an incubator (FOC 225 I, VELP Scientifica,
Usmate Velate MB, Italy) to allow the incorporation of $^{18}O$ into the bacterial DNA. Following
incubation, the microcosms were sacrificed, and each biocrust sample was divided into 4
bead beating tubes (Qiagen, Hilden, Germany), each containing 0.25 g of soil, and stored at -
80 °C until further analysis.
Each labeled sample had a non-labeled control, incubated under identical conditions but
with DNase-free water instead of $^{18}O$ water.

### 2.4.2. DNA extraction

DNA was extracted from all biocrust samples using DNeasy PowerSoil Pro Kit (Qiagen),
according to the manufacturer's instructions. The biomass in hyperarid biocrusts tends to be
very low, yielding only minute amounts of DNA. Therefore, each 1 g soil was extracted in
batches of 0.25 g, and the extracts were later consolidated to increase DNA yield.

### 2.4.3. SIP gradient preparation and fractionation

DNA (ca. 3.5 ng) was subjected to isopycnic gradient centrifugation in a solution of caesium
chloride (7.163 M; CsCl, Sigma Aldrich. St Loise, MI, USA) and buffer (0.1 M Tris-HCl at pH 8.0,
0.1 M KCl and 1 mM EDTA, all from Sigma Aldrich) to a final density of 1.725 g mL$^{-1}$ as described
previously (Jia et al., 2019). The tubes were spun for 44 hr at 177,000 g and then fractionated
by water displacement using a syringe pump (NE-300 Just Infusion™ Syringe Pump, NewEra
Pump systems, Farmingdale, NY, USA). The refractive index was measured using an AR200
digital refractometer (Reichert, Depew, NY, USA) and then the DNA was precipitated using a
Polyethylene Glycol 6000 solution (30% PEG 8000 and 1.6 M NaCl), and 30 µg of GlycoBlue
Coprecipitant (Thermo Fisher Scientific, Waltham, MS, USA). Copy numbers of the 16S rRNA
gene in each fraction were determined by qPCR using a probe-based approach. Primers 338F
and 805R (Yu et al., 2005) coupled with a 516P probe (FAM-BHQ1 dual labeled) were used for
the assay. Per one reaction 10 µL of TaqMan™ Fast Advanced Master Mix (Thermo Fisher
Scientific), 0.4 µL of Bovine Saline Albumin (BSA; Thermo Fisher Scientific), 1 µL of each primer
(10 µM), 0.4 µL of a probe (10 µM) and 2.2 µL of PCR water was combined and mixed with 5 µL
of DNA. After 5 min initial denaturation at 95 °C, cycling program: 40 cycles of 95 °C for 30 sec
followed by 62 °C for 1 min was applied. Gene copy numbers were established from a standard
curve of *Escherichia coli* 16S rRNA gene.

### 174    2.4.4. PCR and sequencing

Following fractionation, all samples (labeled and unlabeled) were amplified using the 16S
rRNA primers 515F_mod and 806R_mod (Apprill et al., 2015; Parada et al., 2016). Each reaction
consisted of 2.5 µL Green Taq Buffer (Thermo Fisher Scientific), 2.5 µL of dNTP set
(Biotechrabbit, Berlin, Germany), 0.1 µL of BSA (Thermo Fisher Scientific), 0.625 µL of each
primer (10 µM), 0.125 µL DreamTaq Green DNA Polymerase (Thermo Fisher Scientific) and 17.5
µL of PCR water (Sigma). The PCR ran for 38 cycles using the following program: denaturation
at 94 °C for 45 sec, annealing at 52 °C for 45 sec, extension at 72 °C for 45 sec, and a final cycle
of extension at 72 °C for 10 min. The amplified fragments were sequenced using MiniSeq
(Illumina, San Diego, CA, USA) at the UIC sequencing core, University of Illinois, Chicago, Illinois
(https://rrc.uic.edu/cores/genome-research/genome-research-core/). DNA extraction and SIP
gradient controls, PCR negative controls and mock community (ZymoBIOMICS Microbial
Community Standard II Log Distribution; Zymo Research, Irvine, CA, USA) samples (2 of each)
were also sequenced to control for contaminants in the sequencing results.

## 2.5. Bioinformatic analysis

All the bioinformatic and statistical analyses were done in R V4.1.1 (R development core
team, 2013). Labeling of bacteria was detected using differential abundance analysis as
described in Angel (2019). Briefly, the sequences were processed using the DADA2 package
V8.8 (Callahan et al., 2016) for quality filtering, denoising, read-merging, chimera removal,
constructing amplicon sequence variants (ASV) tables, and taxonomic assignment. Detection
and removal of potential contaminant sequences were performed using the R package
decontam V.1.12.0 (Davis et al., 2017). Prevalence filtering of rare ASVs was done using the
Phyloseq package V1.36.0 (McMurdie and Holmes, 2013). ASVs that appeared in less than
2.5% of the samples were removed. A maximum-likelihood phylogenetic tree was calculated
using IQ-TREE2 V 2.1.1. (Minh et al., 2020). Finally, differential abundance analysis was
performed using DESeq2 V1.32.0 (Love et al., 2014) to compare the relative abundance of
each ASV in the heavy fractions of labeled DNA to the unlabeled heavy fractions (the negative
control samples), which allows identifying the bacterial groups that incorporated the water
isotope into their DNA. The results were filtered to include only ASVs with a 2-fold log change
and a significance value $p < 0.1$.

## 2.6. Predictions of genomic functions

Abundances of functional genes based on 16S rRNA gene abundances were performed using
PICRUSt2 (Douglas et al., 2019). Abundances of functional genes were predicted based on a
filtered ASV table containing only ASVs belonging to proliferated bacteria based on the
differential abundance modeling. The resulting output is functional identifications that were
annotated using the KEGG database to infer functional gene families. Each gene was then
classified into a function category and the abundance of genes within each category was
averaged. The function categories were chosen based on Meier et al. (2021). In their study,
Meier et al. collected biocrusts from the Negev and analyzed bacterial metagenomes in the
biocrusts to evaluate the distribution of metabolic potential among bacterial populations. To
compare functional potential between various bacterial phyla, they selected metabolic genes
encoded in the metagenomic-assembled genomes and grouped them into 10 function
categories.

## 220 2.7. Statistical analyses

chlorophyll *a* concentrations were visualized as an estimation plot using the dabestr
package V0.3.0 (Ho and Tumkaya, 2018). The effect size was calculated as a bootstrap 95%
confidence interval. Relative abundances of phyla, Abundances of functional genes and soil
properties were compared between natural and post-mining biocrusts using Mann-Whitney
tests. The community composition of natural and post-mining biocrusts was assessed using
only sequences belonging to proliferated bacteria based on DESeq2 modeling. The weighted
UniFrac (Lozupone et al., 2011) was used to calculate the similarity between the natural and
post-mining communities, and an adonis model was used to assess whether communities
differ significantly from each other (package Vegan V2.6-2; Dixon, 2003).

## 3. Results

### Sample wetting and greening

Most biocrust samples (both natural and post-mining) showed greening within 36 to 48 hr
into the 96 hr incubation. By the end, most samples displayed varying degrees of greening,
indicating cyanobacterial activity. Generally, post-mining biocrust showed less greening than
the natural biocrusts (Fig. 2).

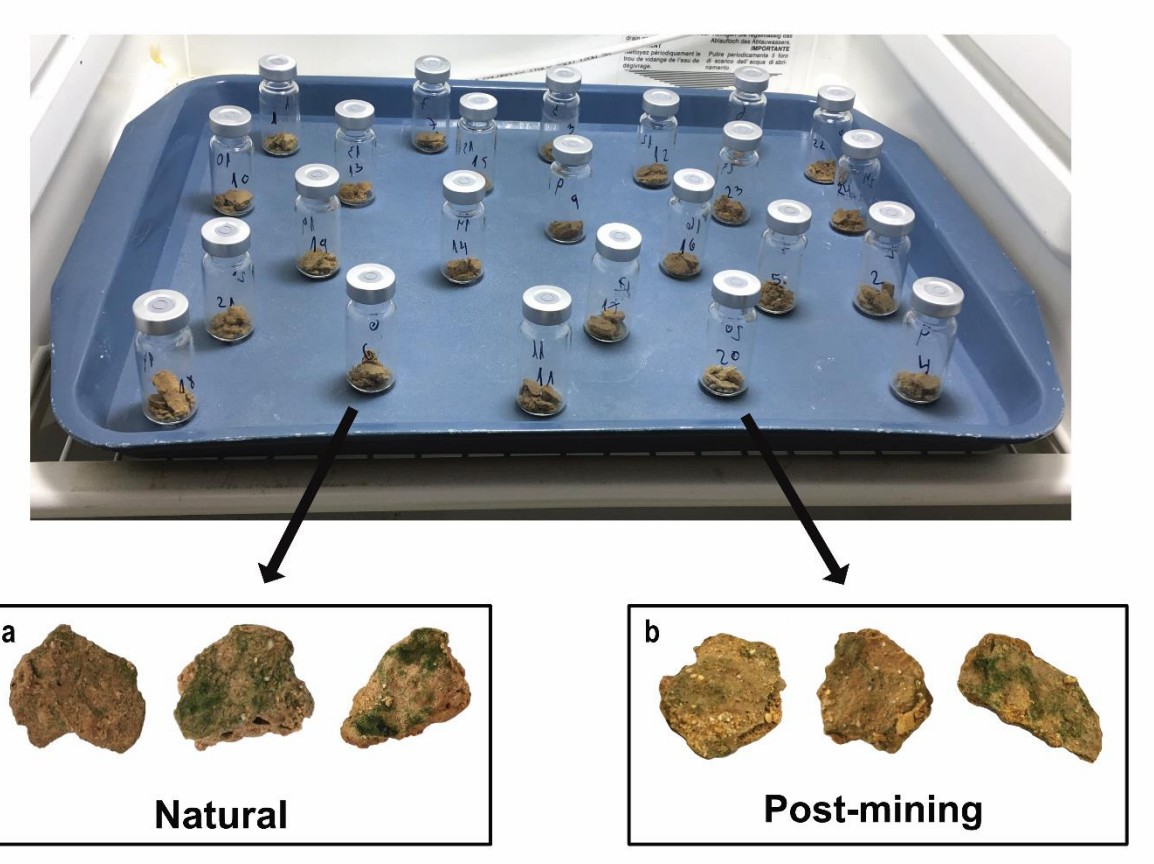

Figure 2: Incubation setup. Top picture – biocrusts in sealed glass vials in the incubator.
Bottom picture – natural (a) and post-mining (b) biocrusts following the 96-hour incubation.

## Soil properties

EC and $NO_3^-$ were significantly higher in natural biocrusts compared to post-mining biocrusts (EC: t = 2.89, p < 0.05; $NO_3^-$: t = 4, p < 0.01; Table 1). Soil organic matter was also significantly higher in the natural biocrusts (t = 3.77, p < 0.01; Table 1). pH was slightly higher in natural biocrusts; however, the differences were not statistically significant (pH: t = 1.41, p = 0.19; Table 1).

Table 1: soil properties for natural and post-mining biocrusts. The numbers represent the means for each property and the standard deviation. Significant differences are marked with an asterisk (* = p < 0.05; ** = p < 0.01).

| Plot type/Soil property | Natural | Post-mining |
|---|---|---|
| pH | 7.6 ± 0.12 | 7.5 ± 0.1 |
| EC | 26.22* ± 9.38 | 9.94 ± 8.39 |
| NO3 | 84.82** ± 36.69 | 14.75 ± 13.57 |
| Soil organic matter | 1.2** ± 0.19 | 0.81 ± 0.12 |

## Chlorophyll *a*

The estimation plot revealed an effect size estimate at 1.42 (95CI -0.432; 3.03; Fig. 3). In the natural samples, there was no clear clustering according to the soil water content i.e., dry or hydrated (following 96 hr incubation with water). In fact, there was a larger variance between samples collected after incubation (Fig. 3). Hydrated post-mining biocrusts had consistently higher chlorophyll *a* concentrations compared to dry biocrusts. It Is also apparent that the variance between samples was smaller in the post-mining biocrusts (Fig. 3).

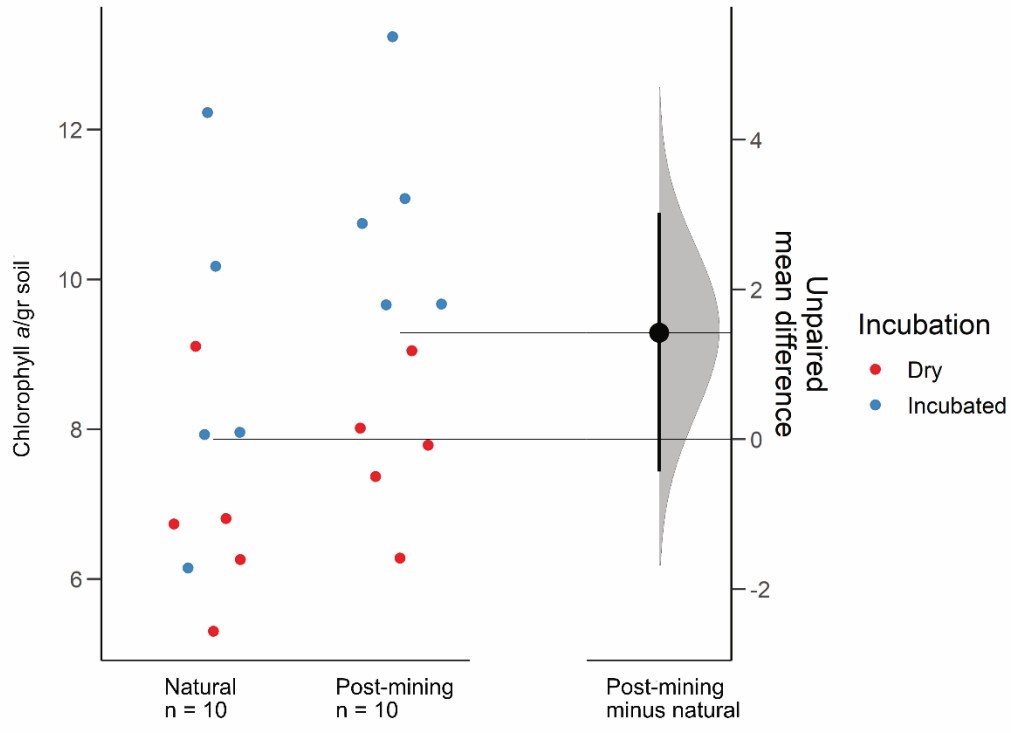

Figure 3: Estimation plots of chlorophyll *a* concentrations. Dots represent the biocrust
samples, and colors represent either dry or incubated soil.

## Sequencing and differential abundance modeling

Sequencing resulted in 47,311 reads per sample on average (Table S2) and 10,275 ASVs
(Table S3). Following decontamination and filtering, 86% of the ASVs were removed (Table
S3). However, they accounted for only 16% of the total reads. Out of the remaining 1,404
ASVs, 1,266 in total were labeled and used for the differential abundance modeling (Table
S3). Each sequence in the labeled samples was compared to its corresponding negative
control, and the $Log_2$-fold change in labeled sequences was evaluated to determine whether
an ASV could be considered truly labeled (i.e., belonging to growing bacteria) based on the
significance threshold. One of the natural biocrust samples (no. 1; Fig. 4) displayed much
higher labeling than the other 4 samples (414 ASVs passed, out of a total of 1,093; Fig. 4).
Excluding sample 1, 38 out of 975 ASVs total passed the significance threshold for $Log_2$ fold
change. In post-mining samples, the number of labeled reads was more consistent among
the different samples (Fig. 4). 68 ASVs out of 874 ASVs total passed the threshold for $Log_2$ fold
change. Altogether, the number of labeled ASVs did not differ significantly between natural
and post-mining samples (natural sample 1 was excluded, natural community mean = 9.5,
post-mining community mean = 13.6, W= 9, p = 0.9).

**(a)**

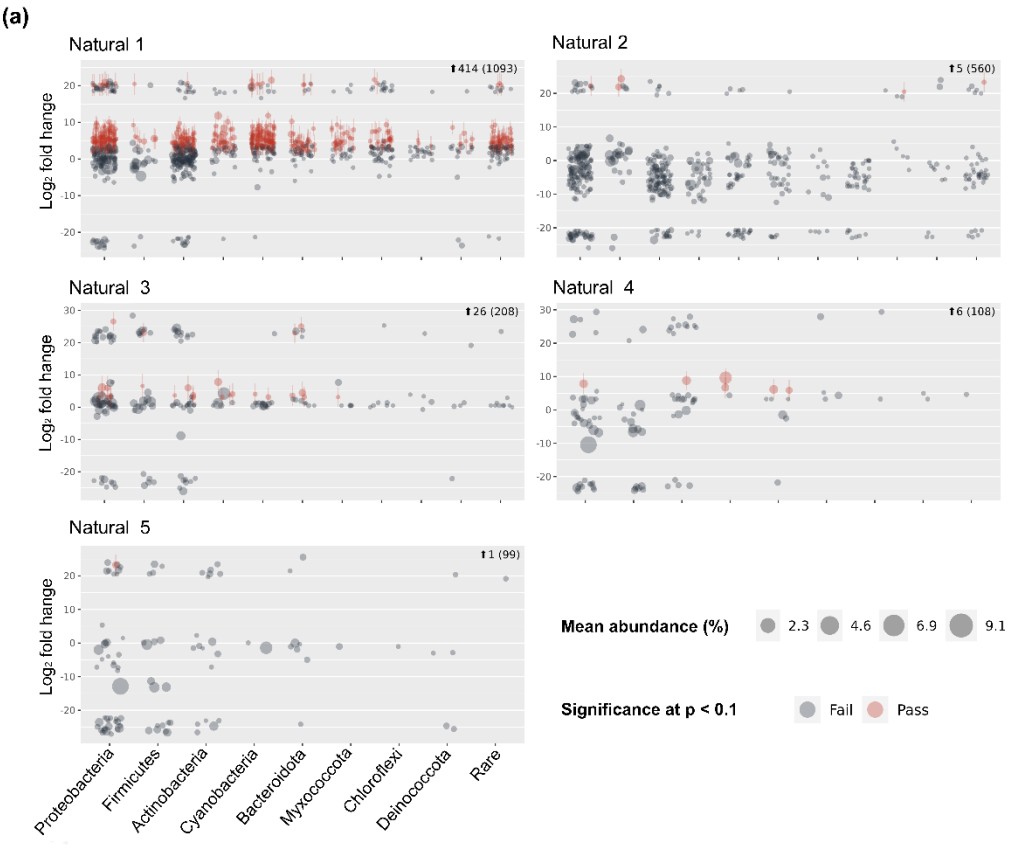

**(b)**

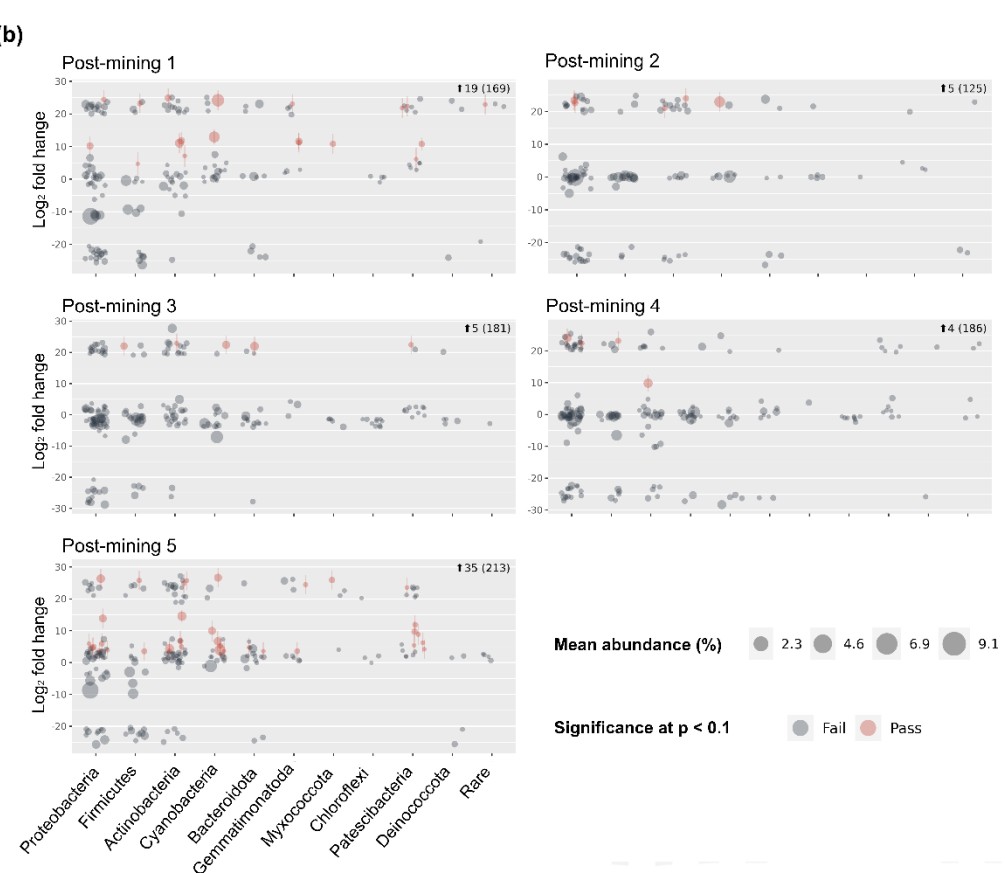


Figure 4: community composition of proliferated bacteria in natural (a) and post-mining (b)
biocrusts. Each graph represents a different sample. Red dots indicate labeled ASVs, and
grey dots indicate unlabeled ASVs, based on Deseq2 modeling.

## Composition of the proliferated bacterial community

Figure 5(a) depicts a PCoA ordination based on weighted UniFrac metric showing that the
biocrust samples do not cluster according to plot type (natural sample number 1 was
excluded). Furthermore, the adonis test revealed no significant differences in community
composition (Weighted UniFrac ~ Plot type; F = 1.23, $R^2$ = 0.15, p = 0.21). A comparison of
phyla relative abundances reveals higher abundances of Cyanobacteria and Actinobacteria
in post-mining samples, and higher abundances of Firmicutes and Proteobacteria in natural
samples (Fig. 5(b)). However, none of the abundances differ significantly between groups
(Table S4). A Venn diagram of unique and overlapping sequences reveals that only 8 out of
88 labeled sequences appear both in natural and post-mining samples (Fig. S2). However,
phylogenetic trees depicting the different proliferated bacterial groups indicate that, for the
most part, sequences that appear in natural and post-mining biocrusts belong to the same
orders/classes. In the phylum Cyanobacteria, labeled sequences belonged to two classes,
and most sequences in both natural and post-mining samples belonged to the class
Cyanobacteria, with a slightly higher prevalence in the post-mining samples (Fig. S1). The
class Bacteroidia, belonging to the phylum Bacteroidota, had a similar prevalence for natural
and post-mining samples (Fig. S1). The trend was similar in the class Bacilli, belonging to the
phylum Firmicutes (Fig. S1). In the Alphaproteobacteria phylum, the orders
Rhodobacteriales, Rhizobiales and Sphingomonadales appeared in both natural and post-
mining samples (Fig. S1). The phylum Gammaproteobacteria appeared only once in post-
mining samples but was more prevalent in natural samples (Fig. S1). The phylum
Actinobacteria was more prevalent in post-mining samples, yet the orders Frankiales,
Micrococcales and Propionibacteriales appeared in both natural and post-mining samples
(Fig. S1).

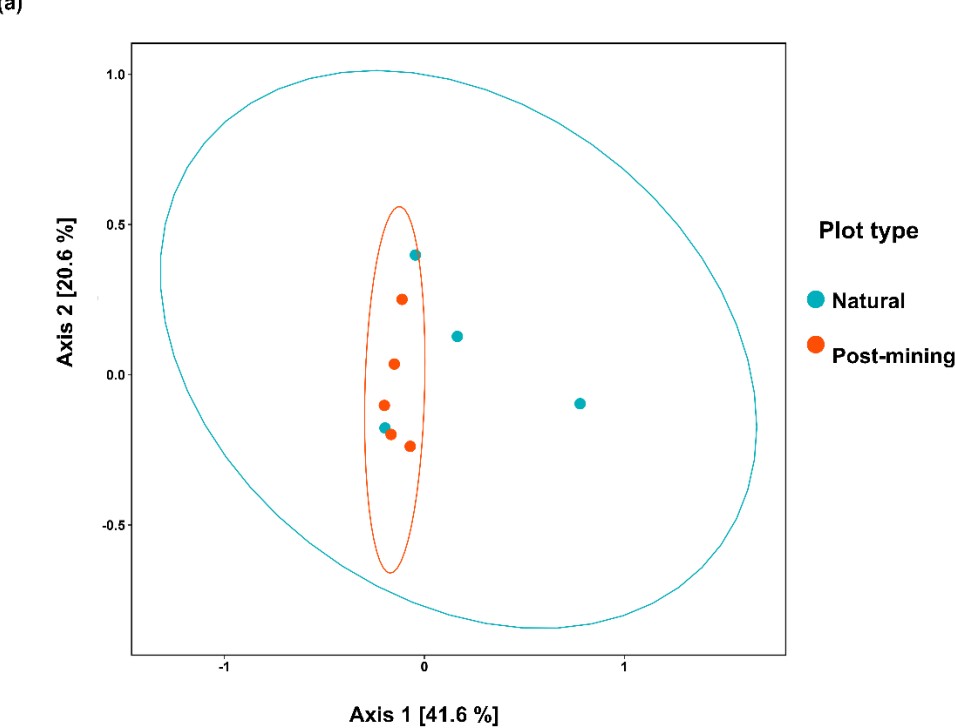

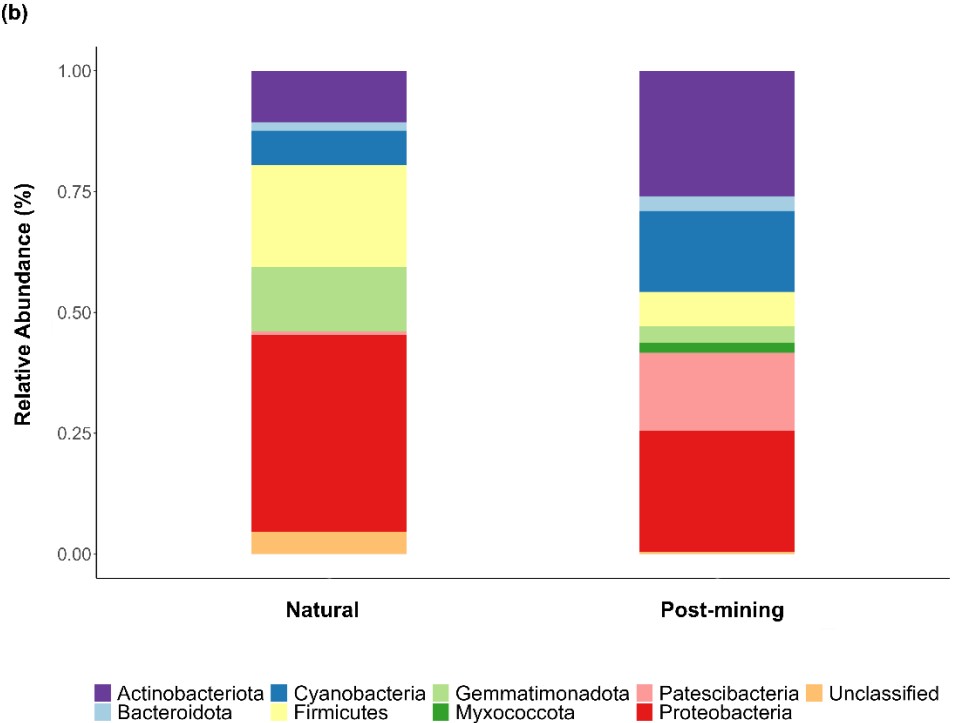


Figure 5: Composition of proliferated community. Top figure (a) depicts a PCoA ordination of community composition based on weighted UniFrac similarity metric. Blue dots are natural samples and pink dots are post-mining samples. The ellipses represent 95% confidence intervals; the bottom figure (b) depicts a bar plot of phyla relative abundance (%) in natural and post-mining biocrusts.

## Predictions of genomic functions


Abundances of 10 function categories (listed in Table S5) were compared between natural
and post-mining biocrust samples. Abundances were generally higher in post-mining
compared to natural biocrusts (Fig. 6; Table S5). Also, the variance between samples was
larger in post-mining biocrust (Fig. 6). However, the differences between plot types were not
statistically significant in any of the function categories (Table S5).

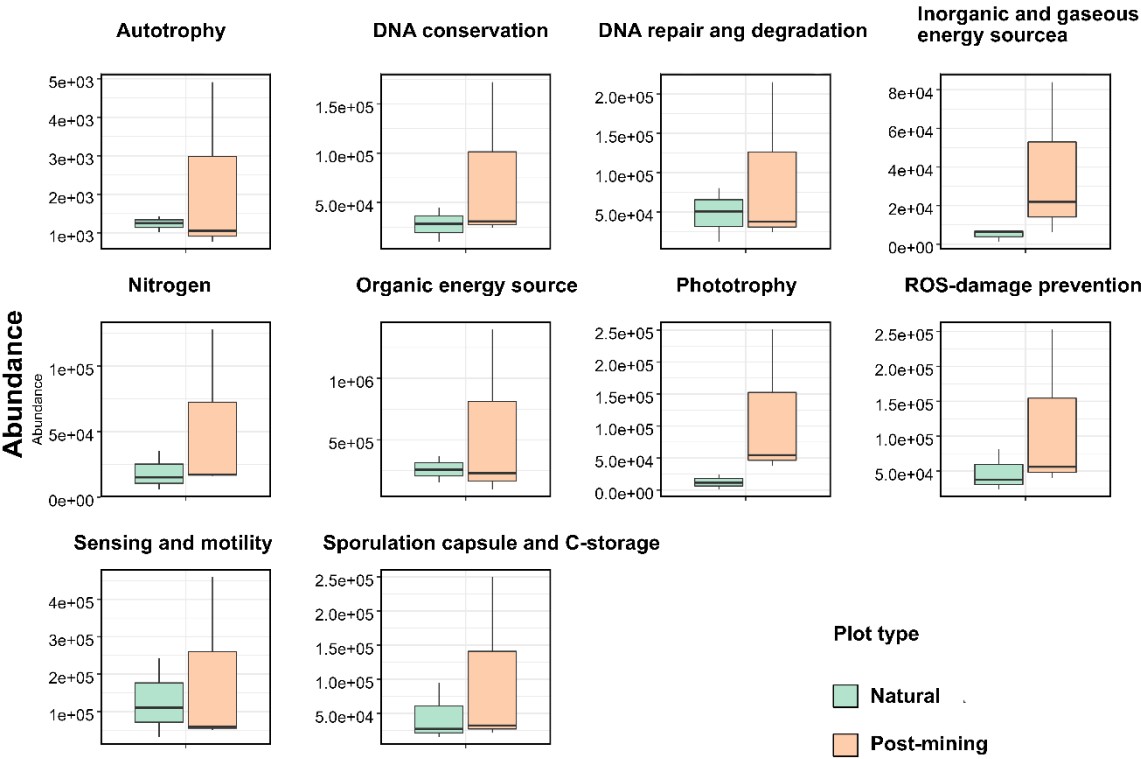


Figure 6: boxplot of functional predictions. The Y axis represents functional gene
abundances. The line represents the median, and the whiskers represent the range.


## Discussion

In this study, we examined which groups of the biocrust bacterial communities grow after hydration using a SIP assay and differential abundance and diversity modelling. We hydrated and incubated the biocrusts for 96 hr expecting bacterial growth, yet very little growth was detected. Only 3.9% of the natural and 7.7% of the post-mining biocrusts' ASVs were identified as truly labeled by the stable isotope. Post-mining biocrusts had a slightly higher number of labeled ASVs compared to natural biocrusts but, the differences were not significant. Also, the composition and taxonomic identity of the growing communities did not significantly differ between natural and post-mining biocrusts.

Biocrust organisms are known to resume activity quickly following hydration, resuming functions such as damage repair, germination, nutrient cycling, and growth (Harel et al., 2004, Rajeev et al., 2013, Green and Proctor, 2016, Thomas et al., 2022). Hydration was also demonstrated to change the biocrust bacterial communities (Angel and Conrad, 2013; Štovícek and Gillor, 2022). In a $H_2^{18}O$ SIP assay using the Negev Desert biocrusts from arid and hyperarid regions, samples were hydrated and incubated for three weeks at maximum water holding capacity. Within days, changes in the labeled bacterial community composition and abundance were observed, indicating growth (Angel and Conrad, 2013). Similarly, biocrusts collected in the Negev Desert Highlands during a rain event and subsequent desiccation, demonstrated an increase in Cyanobacteria and decrease in Actinobacteria relative

abundance  (Baubin et al., 2021), implying selective activation of bacterial taxa in the
hydrated biocrust.

In other $H_2^{18}O$ SIP assays on soil bacterial communities, a quick response to re-wetting was
observed, and bacterial growth was evident within 24 to 72 hours of incubation (Blazewicz et
al., 2014, Aanderud et al., 2015). Thus, we assumed that hydration and incubation of
hyperarid biocrusts under favorable conditions would result in growth. Previous studies
examining the effect of a physical disturbance (repeated trampling) on biocrust
communities, revealed a decrease in the amount of extractable DNA, lower chlorophyll *a*,
and a decrease in biomass and cyanobacteria abundance (Kuske et al., 2012; Steven et al.,
2015; Chung et al., 2019). However, these studies investigated a localized disturbance
compared to mining disturbance, where the biocrust is completely removed over large
spatial scales. Moreover, the previous studies were conducted in environments that were
less extreme than the hyper-arid Zin Valley. Therefore, we expected the damage to the
biocrust in Zin post-mining sites to follow similar patterns but to be more conspicuous than
the previously reported disturbed biocrusts (Kuske et al., 2012; Steven et al., 2015; Chung et
al., 2019). Our previous report (Gabay et al., 2022) supported this notion; we demonstrated
differences in bacterial communities in natural and post-mining biocrusts, expecting these
differences to be reflected in the proliferating communities of these biocrusts.

Our previous survey (2017) also revealed significantly lower abundances of cyanobacteria
and chlorophyll *a* concentrations in post-mining biocrusts (Gabay et al., 2022). Out of the
four mining sites surveyed, Gov (which was restored in 2007) showed the most considerable
shift in biocrust community following mining. However, in the current study, we sampled
post-mining biocrusts at a different location in the Gov mining site (~500 m away from the
original plot) due to technical constraints. In the new location we found that the
photosynthetic potential of the biocrust in the post-mining plots did not differ from the
natural biocrust. These results highlight the importance of microenvironments in shaping
the functionality of biocrusts (Garcia-Pichel and Belnap, 1996). The similarities in active
communities and photosynthetic potential could be due to more developed biocrusts in the
new sampling locations compared to the previous ones.

Photosynthetic activity is usually observed in biocrusts within minutes to hours after
hydration by either dew or rain (Harel et al., 2004; Lange, 2003). In our experiment, we
hydrated the biocrusts to capacity and then incubated the samples for 96 hr. During the
incubation, most biocrust samples displayed some degree of greening, with more greening
in the natural biocrusts (Fig. 2). This indicates that the photosynthetic bacteria in the
biocrust were activated upon wetting. Yet, no significant differences were detected between
natural and post-mining biocrusts chlorophyll *a* concentrations (Fig. 3) or abundances of
photosynthesis related genes (Fig. 6). This implies that similar abundances of photosynthetic
bacteria were activated upon wetting in both biocrusts, yet they barely proliferated (Fig. 4).

The PICRUSt analysis revealed no significant differences in the abundances of genes within
any of the function categories examined (Fig. 6). In contrast, a previous study conducted in
the Negev Desert Highlands, examined active bacterial communities during a rain and
subsequent drying (Baubin et al., 2021). The results indicated an increase in genes related to
photosynthesis,  light, and sensing following the rain, while the other function categories did
not vary significantly. We note that the identity and abundances of the functional genes in
the dry biocrusts detected by Baubin et al., (2021) and here (Fig. 6) are similar. Therefore, we
propose that the similarity between the post-mining and natural biocrust communities (Fig.
5) reflect similar functional potential (Fig. 6). However, low abundance of active ASVs were
used to infer the abundances of functional genes, and large variance between samples in
post-mining biocrusts could mask significant differences (Table S5).

The growth patterns of biocrust organisms are affected by local environmental conditions
(Kim and Or, 2017). Zin mining fields are in a hyperarid region, where extreme heat events are
frequent in the summer, and rains are scarce and unpredicted. Moreover, in recent years
there were only two or three rain events during each rainy season (Zin factory meteorological
data). Hydration is the most important factor affecting biocrust organisms' growth rate while
long desiccation periods negatively affect growth (Zaady et al., 2016). Also, salinity levels in
Zin valley soils are high (Table 1; Levi et al., 2021) imposing further stress on the biocrust
community. It is known that in high stress environments, biocrust microorganisms increase
nutrient availability and accumulation by resuming carbon and nitrogen fixation upon
hydration (Aranibar, 2022). The resulting organic carbon and nitrogen compounds can be
consumed during the long desiccation periods (Belnap, 2003; Colesie et al., 2014). One study
examining microbial nitrogen transformations in biocrusts collected from Succulent Karoo
biome in Namibia and South Africa showed that following wetting, nitrogen cycling genes
are expressed in biocrust organisms (Maier et al., 2022).  Another study examining biocrust
samples taken from the Moab Desert in Utah demonstrated a pulse of metabolite release
following controlled wetting (Swenson et al., 2018). Based on these reports, and due to the
extreme conditions in Zin Valley (Levi et al., 2021), we suggest that hyperarid biocrust
communities prioritize functions such as metabolite production, nutrient cycling and
preparation for desiccation over growth.

Natural recovery of biocrusts has been long debated, and is generally estimated to be a slow
process, especially in arid sites that experience very short activity periods for biocrust
development, such as the hyperarid Zin mining site (Kidron et al., 2020; Weber et al., 2016).
The time and trajectory of recovery depend on many factors relating to local climatic
conditions and site properties (Belnap and Lange, 2003). One such factor that greatly affects
establishment and restoration of biocrusts is the proximity, availability, and dispersal timing
of biocrust propagules (Bowker, 2007; Walker et al., 2007). Thus, the low proliferation rates
we observed, particularly in post-mining biocrusts, suggest that restoration processes might
be much slower than previously estimated. The topsoil from a stockpile is used to cover the
mining pits. This soil may not contain a biocrust seed bank because it was probably
destroyed and buried during the mining processes. Further increase in bacterial biomass
might highly depend on the dispersal of biocrust propagules to the site from adjacent
natural areas by wind or water. Our results further emphasize the need for active restoration
measures in the Zin mines. Such measures include soil inoculation with local cyanobacterial
propagules ( Acea, 2003; Wang et al., 2009; Zhao et al., 2016; Velasco Ayuso et al., 2017) and
increased hydration (Morillas and Gallardo, 2015; Zhang et al., 2018) which were effective in
enhancing biocrust establishment and recovery following disturbances (Antoninka et al.,

442    2020).


## Conclusions


Low proliferation of biocrust bacteria was detected after wetting suggesting prolonged
recovery times of biocrusts following major mechanical disturbances, such as mining.
Furthermore, recovery largely depends on site conditions and the ability of biocrust
propagules to disperse to post-mining sites. Further research is needed to confirm our
hypothesis of low proliferation and thus restoration rates in hyper-arid biocrust bacterial
communities.

## Code and data availability

All data produced in this study and scripts used for community analysis, functional predictions and chlorophyll *a* estimation plot are available at https://github.com/TaliaJoanne/SIP-experiment-Zin-mines.

The raw 16S sequences are available in the NCBI database under Bioproject ID PRJNA906925, accession numbers SAMN31937891 – SAMN31937900.

## Author's contributions

Talia Gabay: Conception, Sample Collection, Incubation, Chlorophyll *a* and DNA extraction, Statistical Analysis, Visualization, Writing-Original Draft Preparation, Writing-Reviewing and Editing. Eva Petrova: DNA-SIP assay, DNA quantification and PCR amplification. Osnat Gillor and Yaron Ziv: Conception, Writing-Reviewing and Editing, Investigation, Supervision. Roey Angel: Conception, Statistical Analysis, Visualization, Supervision, Writing-Reviewing and Editing.

All authors read and approved the manuscript.

## Declaration of competing interest

The authors declare that they have no known competing financial interests or personal relationships that could have appeared to influence the work reported in this paper.


## Acknowledgements

We thank Matan Avital from ICL for coordinating sample collection and providing Zin factory
maps and meteorological data, Sharon Moscovitz for her assistance in sample collection and
Ofer Ovadia for suggestions on statistical analyses. Lastly, we thank ICL Rotem LTD for their
support and funding of this research.

## Financial support

Funding for this research was provided by Rotem ICL LTD. RA was supported by the Czech
Science Foundation (Junior Grant No. 19-24309Y), EP was supported by the Czech Ministry of
Education Youth and Sport (EF16_013/0001782 - SoWa Ecosystems Research).

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
