# Peer review of "Only a minority of bacteria grow after wetting in both"

_EGUsphere, 2022_

## Author Response (AR1)

We thank the reviewers for their valuable comments and suggestions that greatly improved our manuscript.

We have responded to all the comments and modified the manuscript accordingly. Below, please find our responses to the comments as well as explicit quotations of main changes we have made.

Anonymous referee #1:

The manuscript "Only a minority of bacteria grow after wetting in both natural and post-mining biocrusts in a hyperarid, phosphate mine" presents the results of a H2-18-O stable isotope probing assay of the growing community members in hyperarid biocrusts that have been disturbed, or not, by mining activities.

Overall, the manuscript is well conceived, written, and presented. The conclusions are sound and justified.

I do have a concern that the authors have missed a swath of the literature that reports the response of biocrusts to physical disturbances, such as chronic physical trampling. These data show that physical disturbances differ from other disturbances and may affect the recovery of the biocrust community. I think some of this literature should be discussed in the context of what these authors see, and how it may be unique in these hyperarid sites.

Response: I added literature regarding physical trampling and how they might compare to the mining disturbance in the hyper-arid Zin Valley (L358 – L370):
…" Previous studies examining the effect of a physical disturbance (repeated trampling) on biocrust communities, revealed a decrease in the amount of extractable DNA, lower chlorophyll a, and a decrease in biomass and cyanobacteria abundance (Kuske et al., 2012; Steven et al., 2015; Chung et al., 2019). However, these studies investigated a localized disturbance compared to mining disturbance, where the biocrust is completely removed over large spatial scales. Moreover, the previous studies were conducted in environments that were less extreme than the hyper-arid Zin Valley. Therefore, we expected the damage to the biocrust in Zin post-mining sites to follow similar patterns but to be more conspicuous than the previously reported disturbed biocrusts (Kuske et al., 2012; Steven et al., 2015; Chung et al., 2019). Our previous report (Gabay et al., 2022) supported this notion; we demonstrated differences in bacterial communities in natural and post-mining biocrusts, expecting these differences to be reflected in the proliferating communities of these biocrusts."

Similarly, transcriptomic studies that show the genes that are expressed during a wetting pulse and have documented metabolic a physiological changes in the bacterial community in a wetting pulse. These studies show where resources may be going to rather than growth, and should probably be at least mentioned in this text.

Response: I added previous studies that showed expression of nitrogen cycling genes and metabolite production following wetting (L417 – L425):
…" One study examining microbial nitrogen transformations in biocrusts collected from Succulent Karoo biome in Namibia and South Africa showed that following wetting, nitrogen cycling genes are expressed in biocrust organisms (Maier et al., 2022). another study examining biocrust samples taken from the Moab Desert in Utah demonstrated a pulse of metabolite release following controlled wetting (Swenson et al., 2018). Based on these reports (Maier et al., 2022; Swenson et al., 2018) and due to the extreme conditions in Zin Valley (Levi et al., 2021), we suggest that hyperarid biocrust communities prioritize functions such as metabolite production, nutrient cycling and preparation for desiccation over growth."

Minor concerns:

Title: I am not sure the comma is needed between hyperarid and phosphate

Response: corrected (removed the comma).

Line 210: picrust should be PICRUSt2

Response: corrected (L208).

Line 214: I think this should be spelled out more completely. It just sends the reader to a reference. This is a specialized set of functional annotations. I think it would help the reader interpret Figure 6 and where the annotations are coming from.

Response: I added a description of the referenced study and explained the source of the functional annotations (L213 – L218):
"…The function categories were chosen based on Meier et al. (2021). In their study, Meier et al. collected biocrusts from the Negev and analyzed bacterial metagenomes in the biocrusts to evaluate the distribution of metabolic potential among bacterial populations. To compare functional potential between various bacterial phyla, they selected metabolic genes encoded in the metagenomic-assembled genomes and grouped them into 10 function categories."

Anonymous referee #2:

This study is interesting as it identifies which bacteria proliferate in biocrusts from natural and postmining areas and examines functional gene abundances, unlike previous studies that have focused on characterizing the microbial community composition of natural and post-mining biocrusts. This, I consider the approach novel.

Nonetheless, I have a few comments that should be addressed:

Abstract

L19-23. I recommend deleting this sentence because it refers to a previous study, and the abstract should focus on the objectives and results of the present study. Then, in the Introduction or the Discussion, it is fine if mention to previous results is done.

Response: I removed the section from the abstract.

M&M

L103-105. It is not clear the predominant particle size in the study site. Were they sandy soils? Also describe briefly the main cover types in the area (bare soil, biocrusts and vegetation?). If existing vegetation, which are the main vegetation species?

Response: I added details regarding soil composition and cover types (L96 – L104):
"…The study area was previously described in Gabay et al., (2022). Briefly, Zin Valley is a hyperarid region of the Negev Desert, with 50 mm average annual rainfall (Zin factory meteorological data) and highly saline soils (average EC = 24 dS/m. The main soil cover types in Zin Valley are biocrusts and desert pavement, with scarce vegetation of mainly annual species. The soil composition in the post-mining site and natural area is similar with 70% sand, 18% silt and 12% clay and 68% sand, 20% silt and 12% clay for natural and post-mining respectively (Gabay et al., 2022). The soils in Zin Valley are classified as Solonchaks according to the World Reference Based soil classification system."

L109. Add a short description of the biocrust sampled (thickness, cover, composition, ….). Were there differences in the biocrust from natural and post-mining sites?

Response: We added a description of the biocrusts in Gov mining site (L106 – L109):
"…Biocrusts were sampled either from the post-mining site or the adjacent natural area. The biocrusts in Gov are thin (between 1.5 - 2 .5 mm deep), and smooth. The site is characterized by areas covered in biocrusts or desert pavement. In each sampling site, we sampled along a 100 m strip at approximately 10 m intervals (Fig. 1)."

Results

Regarding the results of the proliferated bacterial community composition, it would be nice to include a figure with the relative abundances of the different bacteria phylum for natural and post-mining biocrusts to make the results clearer.

Response: I added a bar plot of Phyla relative abundances to the PCoA ordination in figure 5 (labeled 'a' and 'b') and statistical data in the supplementary file (Table S4).

Discussion

L354-357. Can you remind here when was the sampled post-mining site restored? Was it at 2007? I wonder if one reason for the lack of photosynthetic potential difference between post-mining and natural biocrusts is because the biocrust in the post-mining site has been already recovered. For example, soil colonization by cyanobacteria can be faster than thought.

Response: I mentioned the restoration year in L375. I agree that it is likely that the biocrusts we sampled have a significant cyanobacterial community, however we hypothesize that the recovery of the biocrust differs even within the mining site, emphasizing the importance of microenvironments, where certain areas have more developed biocrusts and others have less developed biocrust even within the same post-mining plot. I referenced this in the discussion (L376 – L383):
"…However, in the current study, we sampled post-mining biocrusts at a different location in the Gov mining site (~500 m away from the original plot) due to technical constraints. In the new location we found that the photosynthetic potential of the biocrust in the post-mining plots did not differ from the natural biocrust. These results highlight the importance of microenvironments in shaping the functionality of biocrusts (Garcia-Pichel and Belnap, 1996). The similarities in active communities and photosynthetic potential could be due to more developed biocrusts in the new sampling locations compared to the previous ones."

L369-370. There is little discussion on the results of the Picrust analysis. It would be nice to compare the results of this study with other studies that have also examined functional genes in biocrusts to see if abundances of the functional genes at this site greatly differ from those of biocrusts of other regions.

Response: I added a paragraph addressing the PICRUSt analysis with reference and comparison to a previous study that used PICRUSt to infer functional potential (L395 – L405):
"…The PICRUSt analysis revealed no significant differences in the abundances of genes within any of the function categories examined (Fig. 6). In contrast, a previous study conducted in the Negev Desert Highlands, examined active bacterial communities during a rain and subsequent drying (Baubin et al., 2021). The results indicated an increase in genes related to photosynthesis, light, and sensing following the rain, while the other function categories did not vary significantly. We note that the identity and abundances of the functional genes in the dry biocrusts detected by Baubin et al., (2021) and here (Fig. 6) are similar. Therefore, we propose that the similarity between the post-mining and natural biocrust communities (Fig. 5) reflect similar functional potential (Fig. 6). However, low abundance of active ASVs were used to infer the abundances of functional genes (Fig. 6), and large variance between samples in post-mining biocrusts could mask significant differences (Table S5)."

L379-381. It is hypothesized that the low bacterial growth in harsh environments like this one could be because biocrusts prioritize activation and preparation for desiccation over growth. However, this hypothesis cannot be fully demonstrated with the reported results. Could you add references to support this hypothesis?

Response: Yes, I added references to previous studies that demonstrated functions such as nitrogen cycling and metabolite production following wetting (L417 – L425):
"…One study examining microbial nitrogen transformations in biocrusts collected from Succulent Karoo biome in Namibia and South Africa showed that following wetting, nitrogen cycling genes are expressed in biocrust organisms (Maier et al., 2022).  another study examining biocrust samples taken from the Moab Desert in Utah demonstrated a pulse of metabolite release following controlled wetting (Swenson et al., 2018). Based on these reports (Maier et al., 2022; Swenson et al., 2018) and due to the extreme conditions in Zin Valley (Levi et al., 2021), we suggest that hyperarid

biocrust communities prioritize functions such as metabolite production, nutrient cycling and preparation for desiccation over growth."

L397-401. Is there any practical experience of the application of cyanobacteria inoculation to restore this mining site in Israel? If so, it would be nice to add a few sentences regarding the potential of this technique to restore biocrust in the current mining site.

Response: No, there have not been any active restoration measures in Zin mining sites and no previous scientific studies regarding the mining practices of ICL mining company. I address the need for active restoration measures in Zin mining sites (including cyanobacteria inoculation) in L439 – L444:
"…Our results further emphasize the need for active restoration measures in the Zin mines. Such measures include soil inoculation with local cyanobacterial propagules (Acea, 2003, Wang et al., 2009, Zhao et al., 2016, Velasco Ayuso et al., 2017) and increased hydration (Morillas & Gallardo, 2015,  Zhang et al., 2018), which were effective in enhancing biocrust establishment and recovery following disturbances (Antoninka et al., 2020)."

Table 1. Values should also include the standard deviation.

Response: I added SD values to the table (L254 – L254).